# SUL-150 Limits Vascular Remodeling and Ventricular Failure in Pulmonary Arterial Hypertension

**DOI:** 10.3390/ijms26157181

**Published:** 2025-07-25

**Authors:** Lysanne M. Jorna, Dalibor Nakládal, Johannes N. van Heuveln, Diederik E. van der Feen, Quint A. J. Hagdorn, Guido P. L. Bossers, Annemieke van Oosten, Michel Weij, Ludmila Tkáčiková, Soňa Tkáčiková, Robert H. Henning, Martin C. Harmsen, Rolf M. F. Berger, Guido Krenning

**Affiliations:** 1Department of Pediatric and Congenital Cardiology, University Medical Center Groningen, University of Groningen, Hanzeplein 1 (CA40), 9713 GZ Groningen, The Netherlands; 2Laboratory for Cardiovascular Regenerative Medicine, Department of Pathology and Medical Biology, University Medical Center Groningen, University of Groningen, Hanzeplein 1 (EA11), 9713 GZ Groningen, The Netherlands; 3Department of Clinical Pharmacy and Pharmacology, University Medical Center Groningen, University of Groningen, Hanzeplein 1 (AP50), 9713 GZ Groningen, The Netherlands; 4Clinical Research Unit, Comenius University Science Park, Comenius University Bratislava, 841 04 Bratislava, Slovakia; 5Department of Experimental Pharmacology, Medical Faculty Mannheim, University of Heidelberg, Ludolf Krehlstrasse 13-17, 68167 Mannheim, Germany; 6Department of Microbiology and Immunology, University of Veterinary Medicine and Pharmacy, Komenského 73, 041 81 Košice, Slovakia; 7Department of Medical and Clinical Biophysics, Faculty of Medicine, Pavol Jozef Šafárik University, Trieda SNP 1, 040 11 Košice, Slovakia; 8Sulfateq B.V., Admiraal de Ruyterlaan 5, 9726 GN Groningen, The Netherlands

**Keywords:** pulmonary arterial hypertension (PAH), heart failure, mitochondrial dysfunction, 6-chromanol derivate, SUL-150

## Abstract

Pulmonary arterial hypertension (PAH) is a rare, progressive, and incurable disease characterized by an elevated pulmonary blood pressure, extensive remodeling of the pulmonary vasculature, increased pulmonary vascular resistance, and culminating in right ventricular failure. Mitochondrial dysfunction has a major role in the pathogenesis of PAH and secondary right ventricular failure, and its targeting may offer therapeutic benefit. In this study, we provide proof-of-concept for the use of the mitochondrially active drug SUL-150 to treat PAH. PAH was induced in rats by monocrotaline, followed by the placement of an aortocaval shunt one week later. The mitoprotective compound SUL-150 (~6 mg·kg^−1^·day^−1^) or vehicle was administered intraperitoneally via osmotic minipump for 28 days, implanted at the time of aortocaval shunt placement. Vehicle-treated PAH rats had dyspnea and showed pulmonary artery remodeling with increased responsiveness to phenylephrine, in addition to remodeling of the intrapulmonary arterioles. SUL-150 administration mitigated the dyspnea and the remodeling responses. Vehicle-treated PAH rats developed right ventricular hypertrophy, fibrosis, and failure. SUL-150 administration precluded cardiomyocyte hypertrophy and inhibited ventricular fibrogenesis. Right ventricular failure in vehicle-treated PAH rats induced mitochondrial loss and dysfunction associated with a decrease in mitophagy. SUL-150 was unable to prevent the mitochondrial loss but improved mitochondrial health in the right ventricle, which culminated in the preservation of right ventricular function. We conclude that SUL-150 improves PAH-associated morbidity by the amelioration of pulmonary vascular remodeling and right ventricular failure and may be considered a promising therapeutic candidate to slow disease progression in pulmonary arterial hypertension and secondary right ventricular failure.

## 1. Introduction

Pulmonary arterial hypertension (PAH) is a rare, progressive, and incurable disease characterized by an elevated pulmonary blood pressure and extensive remodeling of the pulmonary vasculature, culminating in widespread endothelial dysfunction, stiffening of the elastic proximal pulmonary arteries, pulmonary vasoconstriction, fibrosis, and the development of vaso-occlusive lesions [1]. Globally, a rough estimate of 15 patients per million population suffer from PAH [2], who have a 5-year survival rate ranging from 45–65% [3]. Right ventricular failure due to its increased afterload is the main contributor to mortality in PAH patients [4].

Current therapies for PAH focus on improving endothelial cell function and reducing pulmonary vascular pressure and remodeling [5]. This is achieved by targeting NO deficiencies (PDE-type 5 inhibition), vasoconstriction (prostacyclin analogues, Ca^2+^-channel blockers, or endothelin receptor antagonists), or decreasing the circulatory volume (diuretics) [5,6]. Albeit these therapies have been shown to improve patient quality of life, no PAH therapies exist to date that mitigate disease progression, and lung transplantation remains the only viable option for patients not responding to therapy [6].

Recent evidence suggests that mitochondrial dysfunction has a major role in the pathogenesis of PAH and secondary right ventricular failure, wherein mitochondrial respiration and redox signaling are dysfunctional [7,8]. In the pulmonary vasculature, mitochondrial dysfunction contributes to the formation of excessive reactive oxygen species and the scavenging of NO and facilitates smooth muscle cell proliferation and contraction, and thus increased pulmonary vasoconstriction [9,10]. In the right ventricle, mitochondrial dysfunction contributes to ventricular hypertrophy and fibrogenesis and impairs right ventricular contraction [8,11]. Combined, these data suggest that mitochondrial dysfunction is a plausible therapeutic target in the treatment of PAH and secondary right ventricular failure. Indeed, reduced mitochondrial respiration is associated with PAH [12], which may be ameliorated by increasing glucose oxidation [13], and oral supplementation with the mitochondrially active coenzyme Q modestly increases cardiac function in PAH patients [14]. Yet, no mitochondrially targeted drugs are available for PAH.

A new class of 6-chromanol-based compounds was developed to preserve mitochondrial respiration by the activation of complex I and complex IV of the mitochondrial respiration chain [15], and to maintain mitochondrial ATP production and preclude mitochondrial ROS formation under pathophysiological conditions such as hypothermia, inflammation, and metabolic stress [16,17,18]. Of interest, next to the preservation of mitochondrial function, the 6-chromanol SUL-150 lowers arterial pressure, in part by its function as an α_1_-adrenergic receptor antagonist [19]. Hence, we hypothesize that the administration of SUL-150 slows down disease progression by the amelioration of mitochondrial dysfunction and the reduction in pulmonary blood pressure during PAH pathogenesis.

Here, we provide proof-of-concept for the amelioration of the PAH-associated cardiac and vascular pathophysiology in rats by the mitochondrially active compound SUL-150.

## 2. Results

### 2.1. Development of PAH and the Reduction in Phenylephrine-Induced Constriction of the Pulmonary Artery by SUL-150

Experimental PAH was induced by monocrotaline (MCT) administration and the placement of an aortocaval (AC) shunt. At day 21 post-MCT administration, all animals had developed PAH as indicated by an increased mean pulmonary arterial blood pressure (mPAP) as compared to sham control rats (31.9 ± 1.8 versus 20.2 ± 1.8 mmHg, respectively; *p* < 0.05; Table 1).

Ex vivo, PAH did not affect the potency (EC_50_) and efficacy (E_max_) of phenylephrine-induced vasoconstriction in pulmonary arteries compared to sham control rats (Figure 1). In pulmonary artery rings from sham rats, preincubation with SUL-150 reduced the efficacy of phenylephrine-induced vasoconstriction (15.6 versus 34.7% of max, respectively; *p* = 0.043), but not its potency (9.36 × 10^−7^ versus 1.74 × 10^−7^ M phenylephrine) as compared to vehicle controls (Figure 1). In the pulmonary artery rings from PAH rats, SUL-150 pre-incubation increased the EC_50_ from 1.08 × 10^−6^ to 5.04 × 10^−6^ M phenylephrine (*p* = 0.002) and decreased the E_max_ from 43.4 to 11.0% of max (*p* = 0.005; Figure 1), indicating that SUL-150 decreases both the efficacy and potency of phenylephrine-induced pulmonary artery constriction ex vivo.

### 2.2. Development of PAH and the Reduction in Animal Discomfort by SUL-150

Next, we assessed the effects of chronic SUL-150 administration on the pathophysiology of PAH, using rats implanted with an intraperitoneal osmotic minipump administering vehicle (DMSO) or ~6.0 mg·kg^−1^·day^−1^ SUL-150. Compound administration was successful in the included rats, and the mean concentrations of SUL-150 in blood plasma, lung, and right ventricle were 30.33 ± 34.09 ng·mL^−1^, 27.63 ± 15.77 pg·mg^−1^ and 41.55 ± 50.35 pg·mg^−1^, respectively (see Table 2 for individual values), at the time of sacrifice.

All rats administered with MCT and receiving AC shunting developed PAH with an elevated mPAP as compared to sham rats (31.3 mmHg for vehicle-treated rats and 27.7 for SUL-150-treated rats versus 21.6 mmHg for sham control rats, respectively; *p* < 0.05, Table 3). Rats with experimental PAH and vehicle treatment displayed signs of general discomfort, i.e., a reduction in body weight gain (31.5 ± 6.6 versus 15.2 ± 5.0%—gain for sham and vehicle-treated PAH rats, respectively; *p* < 0.0001), which was not altered by SUL-150 treatment (Table 4). In addition, rats with PAH that received vehicle treatment suffered from dyspnea (*n* = 6, 85.7%), cyanosis (*n* = 1, 14.3%), and edema (*n* = 1, 14.3%). SUL-150 treatment decreased the frequency of dyspnea (n = 2, 33.3%; *p* = 0.040) and edema (0%; *p* < 0.0001). Expectedly, SUL-150 did not affect cyanosis in PAH rats (Table 4). Interestingly, SUL-150 treatment precluded mortality of PAH rats, having a 100% survival rate. (Table 4).

### 2.3. SUL-150 Ameliorates Pulmonary Artery Hemodynamics by the Reduction in Pulmonary Artery Remodeling

Pulmonary artery pressures (Figure 2A and Table 3) were increased in vehicle-treated PAH rats, with an mPAP rising from 21.6 ± 2.1 in vehicle-treated sham control rats to 31.3 ± 7.1 mmHg in PAH rats. Chronic SUL-150 administration tended to decrease mPAP to 27.7 ± 5.8 mmHg, albeit not statistically significant (Figure 2A). Of note, the decrease in mPAP was primarily caused by a lower diastolic pulmonary artery pressure (25% decrease compared to PAH rats), whereas systolic pulmonary artery pressure remained as high as in vehicle-treated PAH rats (Table 3). Pulmonary vascular resistance was increased in vehicle-treated PAH rats, rising from 0.08 ± 0.01 in vehicle-treated sham control rats to 0.30 ± 0.15 mmHg·ml·min^−1^ in PAH rats (Table 3). Chronic SUL-150 administration normalized the vascular resistance to 0.12 ± 0.05 (*p* < 0.05 versus vehicle-treated PAH rats and non-significant versus sham control rats, Table 3).

In ex vivo pulmonary arterial rings, phenylephrine vasoconstrictive potency was decreased in vehicle-treated PAH rats as compared to sham control rats (EC_50_ is 1.24 × 10^−6^ and 8.98 × 10^−8^ M phenylephrine, respectively; *p* = 0.031), while the efficacy of phenylephrine-induced vasoconstriction was increased (E_max_ is 71.9 ± 9.2 and 33.0 ± 2.3% of max, respectively; *p* = 0.011; Figure 2B). Chronic SUL-150 administration in PAH rats reduced the efficacy of phenylephrine-induced vasoconstriction to the level of sham control rats (E_max_ is 33.3 ± 4.7 for SUL-150-treated rats and 33.0 ± 2.3% of max for *p* = 0.984; Figure 2B), without effect on potency (EC_50_ is 1.24 × 10^−6^ for vehicle-treated PAH rats and 1.09 × 10^−6^ M phenylephrine for SUL-150-treated rats, *p* = 0.473).

The thickness of the pulmonary wall was increased in PAH rats as compared to sham control rats (140.0 ± 42.3 and 37.9 ± 10.0 µm, respectively; Figure 2C,D; *p* < 0.001), suggesting that increased muscularization may underlie the increased responsiveness to phenylephrine. Chronic SUL-150 administration substantially reduced the PAH-provoked increase in pulmonary artery wall thickness, measuring 57.6 ± 13.0 µm (Figure 2C,D).

### 2.4. SUL-150 Reduces Small Pulmonary Vessel Remodeling During Pulmonary Arterial Hypertension

PAH is associated with a remodeling response of the intra-acinar pulmonary vessels (Figure 3A), resulting in increased pulmonary resistance (Table 3) and decreased pulmonary perfusion. Corroboratively, PAH rats had a decreased cross-sectional luminal area compared to sham control rats (307.9 ± 58.7 and 586.4 ± 78.3 µm^2^, respectively; *p* < 0.0001; Figure 3B) and an increased occlusion score (46.2 ± 5.9 and 22.9 ± 1.5%, respectively; *p* < 0.0001; Figure 3C). Chronic SUL-150 administration in PAH rats tended to increase cross-sectional lumen area (to 365.7 ± 101.1 µm^2^; *p* = 0.21). The intra-acinar vessel occlusion score decreased upon chronic treatment from 46.2 ± 5.9 for vehicle-treated rats to 39.0 ± 7.4% (*p* = 0.029; Figure 3C), which collectively may explain the decrease in pulmonary vascular resistance.

The vascular remodeling was of the inward-remodeling type, as the cross-sectional vascular area did not differ between groups (957.4 ± 129.3, 986.5 ± 206.8 and 888.2 ± 146.7 µm^2^ for sham, vehicle-treated PAH, and SUL-150-treated PAH rats, respectively), and the wall–lumen ratio increased from 0.16 ± 0.03 for sham rats to 0.62 ± 0.25 µm_wall_·µm_lumen_^−1^ for PAH rats (*p* < 0.0001; Figure 3D). SUL-150 treatment decreased the wall–lumen ratio in PAH rats from to 0.42 ± 0.12 µm_wall_·µm_lumen_^−1^ (*p* = 0.039; Figure 3D). Of note, the percentage of non-perfusable vessels—defined as vessels with a luminal area < 40 µm^2^—was increased in PAH rats compared to sham rats (11.8 ± 7.0 and 0.71 ± 1.22%, respectively; *p* = 0.0001; Figure 3E), and SUL-150 decreased the percentage of non-perfusable vessels to 4.45 ± 2.62% (*p* = 0.009 versus PAH rats; Figure 3E).

### 2.5. SUL-150 Mitigates Right Ventricular Failure Secondary to Pulmonary Arterial Hypertension

Right ventricular failure developed secondary to PAH as indicated by the decreased heart rate (26% decrease) and cardiac output (38% decrease) of vehicle-treated PAH rats as compared to sham rats, and these parameters were both normalized by SUL-150 administration (Figure 4A,B). Right ventricular failure was associated with cardiac hypertrophy, as total heart weight (78%), right ventricular weight (158%), and Fulton index (86%) were increased in vehicle-treated PAH rats compared to sham controls, while eccentricity indexes were decreased (28% systolic and 32% diastolic, respectively; Table 5).

SUL-150 treatment normalized cardiac output (Figure 4B), right ventricle inner diameter in diastole (38% decrease versus PAH rats; *p* = 0.028; Table 5), and right ventricular pressure at diastole (52% decrease versus PAH rats; *p* = 0.005; Figure 4D) to the level of sham control rats, and SUL-150 treatment increased stroke volume by 68% (*p* = 0.001) compared to the stroke volume in vehicle-treated PAH rats (Table 5). Interestingly, the normalization of cardiac output by SUL-150 (Figure 4B) was only modestly reflected by the indicators of cardiac hypertrophy—only the eccentricity indexes were normalized by SUL-150 treatment (Table 5)—which suggests that decreased right ventricular pressure was the major contributor to this effect.

At the tissue level, cardiac hypertrophy was paralleled by cardiomyocyte hypertrophy, shown by an increased cross-sectional area from 180.5 ± 54.7 µm^2^ in sham control rats to 350.3 ± 62.0 µm^2^ in vehicle-treated PAH rats (*p* < 0.0001, Figure 4E,F). Chronic administration of SUL-150 reduced cardiomyocyte size to 246.1 ± 39.4 µm^2^ (*p* = 0.010 versus vehicle-treated PAH rats, Figure 4E,F), which was slightly larger than cardiomyocyte size in sham animals (*p* = 0.0122; Figure 4E,F). To dissect if the decrease in cardiomyocyte hypertrophy resulted from SUL-150 treatment or was secondary to a reduction in pulmonary vascular resistance, we employed an in vitro hypertrophy reporter assay in phenylephrine-treated rat neonatal cardiomyocytes. Phenylephrine stimulation dose-dependently increased the activity of the hypertrophy reporter, which was effectively inhibited by co-treatment with SUL-150 (Figure 4G) and characterized by a 40% reduction in phenylephrine efficacy (E_max_ is 3.93 and 2.36 RFU for vehicle and SUL-150 treatment, respectively; *p* = 0.0003) without an effect on potency (EC_50_ is 1.52 × 10^−6^ and 3.35 × 10^−6^ M phenylephrine for vehicle and SUL-150 treatment, respectively; *p* = 0.379). These data suggest that the mitigation of cardiomyocyte hypertrophy was a direct effect of SUL-150 administration.

Interstitial fibrosis in the cardiac tissue was increased in vehicle-treated PAH rats compared to sham controls (13.9 versus 9.0% of tissue area for PAH rats and sham controls, respectively; *p* = 0.015). Continuous administration of SUL-150 reduced the level of interstitial fibrosis to 9.3% (*p* = 0.026 versus PAH rats), which was not different than the level of interstitial fibrosis of sham controls (Figure 4H,I).

Thus, these data indicate that right ventricular failure was secondary to PAH. Cardiomyocyte hypertrophy and cardiac interstitial fibrosis contribute to the impairment of cardiac function. SUL-150 mitigated right ventricular failure and maintained normal cardiac function.

### 2.6. Right Ventricular Failure Associates with Mitochondrial Wasting and Dysfunction and Is Ameliorated by SUL-150

Dysregulated mitochondrial biogenesis and mitophagy resulting in the loss of cardiac mitochondrial mass have been associated with the development of ventricular failure [20,21]. Expectedly, cardiomyocyte mitochondrial DNA (mtDNA) copy number was decreased in the right ventricle from vehicle-treated PAH rats compared to sham control rats (566.9 ± 247.4 and 977.1 ± 315.0 for vehicle-treated PAH and sham rats, respectively; *p* = 0.006; Figure 5A), indicating the loss of mitochondria. SUL-150 administration did not protect against the loss of mitochondria, and the mtDNA copy number of SUL-150-treated rats was similar to the mtDNA copy number of vehicle-treated PAH rats (Figure 5A).

Oxidative stress was apparent in the right ventricular tissue of vehicle-treated PAH rats as indicated by a reduction in endogenous radical scavenging activity (59% decrease compared to sham rats; *p* = 0.0001; Figure 5B) and an increase in lipid peroxidation products (76% increase compared to sham rats; *p* = 0.0002; Figure 5C). Chronic SUL-150 administration maintained the radical scavenging activity and simultaneously decreased lipid peroxidation level to those of sham rats (Figure 5B,C).

Adenosine triphosphate (ATP), the end-product of mitochondrial respiration, was greatly reduced in the right ventricles of vehicle-treated PAH rats as compared to sham rats (155.0 ± 30.0 and 299.5 ± 49.9 pmol·mg^−1^, respectively; *p* < 0.0001; Figure 5D). SUL-150 administration increased the ATP level of the right ventricle to 272.2 ± 26.6 pmol·mg^−1^ (*p* < 0.0001), which was similar to the level in sham rats (Figure 5D).

The right ventricular expression of molecular markers for mitogenesis, i.e., PGC1α and mtTFA, did not differ between sham control rats and vehicle-treated PAH rats. Also, SUL-150 administration to PAH rats did not alter the protein expression of PGC1α and mtTFA (Figure 5E,F). The right ventricular protein expression of molecular markers of mitophagy, i.e., PINK1 and Parkin, was greatly decreased in vehicle-treated PAH rats as compared to sham rats (3.5-fold and 1.7-fold for PINK1 and Parkin, respectively; Figure 5E,F). SUL-150 administration in PAH rats maintained the expression of both PINK1 and Parkin, and their expression levels did not differ from sham control rats.

Collectively, these data demonstrate that although mitochondrial copy numbers were reduced in both vehicle- and SUL-150-treated PAH rats as compared to sham control rats, SUL-150 administration precludes the development of right ventricular oxidative stress and increases mitochondrial function. Maintenance of mitophagy and thereby mitochondrial health is suggested as the underlying mechanism that is induced by chronic SUL-150 administration.

## 3. Discussion

In the present study, we found that the novel 6-chromanol SUL-150 ameliorates dyspnea, pulmonary vascular remodeling, and right ventricular failure in experimental PAH. We employed a rat model of PAH induced by i.p. MCT administration and followed by AC shunt. We demonstrate that pre-emptive SUL-150 therapy decreases vascular remodeling of the large and small pulmonary arteries, decreases mPAP, and decreases PVR in PAH rats. Moreover, pre-emptive SUL-150 therapy maintains cardiac output, precludes cardiac and cardiomyocyte hypertrophy, and diminishes mitochondrial dysfunction and cardiac oxidative stress in rats with PAH. Given the paucity of therapeutic options that slow down or even halt the progression of PAH and secondary right ventricular failure, SUL-150 emerges as a promising candidate.

The novel 6-chromanol SUL-150 is both mitochondrially active and holds weak α_1_-adrenergic receptor antagonistic properties [19]. Indeed, SUL-150 competitively antagonized phenylephrine-induced vasoconstriction when administered in ex vivo myography and reduced phenylephrine-induced contractile responses after chronic exposure. In PAH rats, the potency of phenylephrine to constrict the pulmonary arteries ex vivo was not different between vehicle- and SUL-150-treated rats, suggesting that chronic SUL-150 administration did not alter adrenoceptor abundance but rather changed the remodeling of the pulmonary vessels. Indeed, chronic SUL-150 administration reduces pulmonary artery remodeling and decreases pulmonary vascular resistance, albeit the underlying mechanism is not identified. Intriguingly, chronic α_1_-adrenergic receptor antagonism [22] and the preclusion of pulmonary artery mitochondrial dysfunction [23] can both decrease pulmonary artery remodeling and thus reduce the vasoconstrictive responses against phenylephrine.

Pulmonary vascular remodeling is the key structural alteration in PAH [24], and the human PAH pathophysiological phenotype is well represented in the rat model of monocrotaline followed by aortocaval shunt [25]. Remodeling of the pulmonary vessels was reduced by SUL-150, as indicated by a reduction in occlusion score, wall/lumen ratio, and the number of non-perfusable vessels. As these parameters contribute to increased vascular resistance in PAH rats [26,27], expectedly, chronic SUL-150 administration decreased their pulmonary vascular resistance.

The development of right ventricular failure secondary to PAH has been attributed to increased pulmonary artery remodeling [28], increased pulmonary vascular resistance [29], and ventricular mitochondrial dysfunction [30], and SUL-150 treatment in PAH rats normalized all these parameters. SUL-150 treatment maintained mitochondrial function and right ventricular ATP production, which may have improved ventricular contractility and thus directly influenced cardiac function. Indeed, decreased ventricular ATP levels are associated with the development of diastolic dysfunction [31] in PAH patients [32] as well as in our rat model. Expectedly, chronic SUL-150 administration in PAH rats normalized cardiac output, which decreased in vehicle-treated PAH rats.

SUL-150 administration did not preclude the loss of right ventricular mitochondria in PAH rats as indicated by a reduction in mtDNA. SUL-150 did, however, efficiently improve mitochondrial health as indicated by enhanced ventricular radical scavenging activity, a decrease in right ventricular peroxidation products and enhanced ATP production. The inability of SUL-150 to preclude mitochondrial loss may be inherent to the timing of its administration. In our model, PAH was induced by a single i.p. monocrotaline injection followed by the placement of an AC shunt 7 days post-monocrotaline administration. SUL-150 therapy also started 7 days post-monocrotaline administration. It is tempting to speculate that the loss of mitochondria was induced by monocrotaline—as monocrotaline is known to induce mitochondrial loss [33]—and had already taken place prior to the administration of SUL-150. Yet, data on the kinetics of mitochondrial loss by monocrotaline are scarce if not non-existent [34].

We investigated mitochondrial biogenesis and mitophagy based on the expression of PGC1α, mtTFA, PINK1, and Parkin—markers of mitogenesis [35,36] and mitophagy [37,38], respectively. In right ventricle homogenates, we found no difference in the protein expression of PGC1α and mtTFA between vehicle and SUL-150-treated PAH rats and sham control rats, suggesting that mitochondrial biogenesis is unaltered during PAH and not affected upon SUL-150 administration. This observation contrasts with earlier reports that describe a marginal decrease in PGC1α and mtTFA during decompensated ventricular dysfunction [39,40]. SUL-150 normalized the expression of PINK1 and Parkin in PAH rats. Concurrent to pertinent literature [40], in vehicle-treated PAH rats, the expression of both PINK1 and Parkin was drastically reduced as compared to sham control rats, suggestive of defective mitophagy. This loss of mitophagy control would result in the accumulation of damaged mitochondria in the ventricular tissue [41]. SUL-150 treatment maintained the expression of both PINK1 and Parkin, suggesting that the mitophagy pathway was active and capable of removing damaged mitochondria from the right ventricular tissue. Collectively, these observations imply that mitochondrial turnover and quality control is disturbed in PAH and that SUL-150 ameliorates this disturbance, resulting in improved mitochondrial ATP production.

To induce PAH in rats, we made use of an aortocaval shunt on top of i.p. monocrotaline administration. Although the combination of monocrotaline and the AC shunt reflects human PAH pathophysiology better than either alone [42], the AC shunt induces a continuous increase in PAP, which cannot be therapeutically overcome. In this context, it is noticeable that SUL-150 decreased diastolic pulmonary pressure and vessel remodeling in such a PAH model. Consequently, the therapeutic effects of SUL-150 may be even larger in PAH models that do not rely on this hemodynamic alteration, such as the models of hypoxia-induced PAH [43] or the sole administration of monocrotaline [44].

The present study does not provide an answer to the question of whether the maintenance of right ventricular function originates from the enhanced pulmonary haemodynamics (i.e., the reduced pulmonary vascular remodeling and vascular resistance) or from the compound’s direct effect on the ventricular cardiomyocytes. Moreover, the preclusion of both pulmonary vascular remodeling and the maintenance of ventricular mitochondrial function may be synergistic in the amelioration of the cardiac PAH phenotype. PAH progression and the occurrence of right ventricular failure are both associated with increased pulmonary vascular resistance [45], and medicaments that decrease vascular resistance often show beneficial cardiac effects [46,47]. Hence, it is conceivable that the increased pulmonary perfusion contributed to the improved right ventricular function.

SUL-150 has both mitoprotective properties and α_1_-adrenoceptor antagonizing potential; we cannot fully dissect the mechanism of action, nor exclude synergy in the modulation of the observed therapeutic effect. Both the inhibition of mitochondrial dysfunction [40,48], and the inhibition of α-adrenoceptor activity [49,50,51] hold therapeutic potential in experimental PAH according to pertinent literature. Follow-up studies that compare the therapeutic potential of pure mitoprotective compounds and pure α-adrenoceptor antagonists may answer this question. In our study, the average plasma level of SUL-150 in PAH rats was 30.3 ng∙mL^−1^ (range 3.3–93.2 ng∙mL^−1^), and, considering the low potency of SUL-150 as an α_1_-adrenoceptor antagonist (pA_2_ ≈ 5.4), it is most plausible that the mitoprotective properties of SUL-150 comprise its primary mechanism of action. Nonetheless, and irrespective of its mechanism of action, SUL-150 administration improves pulmonary vascular and right ventricular function in experimental PAH.

### Limitations of the Study

Several limitations in our study are acknowledged. First, in the current proof-of-concept study, we used a single dose of SUL-150 administered intraperitoneally via osmotic minipump. The dose was based on the maximal solubility of SUL-150 in the vehicle and the flow rate of the minipump. Ideally, dose–response studies and monitoring of plasma and tissue drug concentration are performed to assess the optimal therapeutic dosage. Moreover, in the current study we used continuous intraperitoneal delivery of SUL-150, whereas assessing the optimal delivery route and interval would further establish therapeutic relevance. Studies that further expand on the pharmacokinetic and pharmacodynamic profile of SUL-150 are in future perspective. Nonetheless, in this proof-of-concept study, we show that the pharmacological targeting of mitochondrial function has therapeutic merit in PAH.

Second, patients with PAH are currently treated with prostacyclin analogs, PDE5 inhibitors, and/or endothelin receptor antagonists. Although such treatments improve the quality of life of patients [6], current PAH treatments are focused on symptom relief and are not curative. Therefore, it would be of great interest to compare the therapeutic potential of SUL-150 to standard PAH therapies, and the absence of such a comparator treatment in this study should be considered as a limitation of the current study.

Third, there is a sex dimorphism in PAH, wherein PAH is 2–4 times more prevalent in females [52], albeit PAH progression is more aggressive in the male sex [53]. Our study ignored this sex dimorphism, as we addressed the therapeutic efficacy of SUL-150 only in male rats. Future studies should address if SUL-150 is equally effective in both sexes.

Fourth, albeit our study shows that SUL-150 improves mitochondrial health in the right ventricle, in this proof-of-concept study, we did not fully explore the underlying molecular mechanisms, which may extend to enhanced oxidative phosphorylation, improved calcium homeostasis, or increased mitochondrial dynamics, which may be assessed by investigating molecular markers. Also, processes downstream of the regulation of mitochondrial health, i.e., fibrosis pathways and apoptosis and cell survival pathways, were not further explored here. Therefore, and albeit SUL-150 improves mitochondrial health in PAH, the full molecular mechanism needs further elucidation.

## 4. Materials and Methods

### 4.1. Study Design

The goal of this study was to investigate the efficacy of SUL-150 to preclude morbidity and mortality in experimental PAH. We used an established rat model for severe PAH that combines an MCT injection (60 mg·kg^−1^ i.p.) with an aortocaval shunt. SUL-150 or vehicle was administered continuously for 28 days i.p. by osmotic minipump. At termination, cardiac function was assessed by echocardiogram and hemodynamic measurements. Ex vivo myograms of the pulmonary artery were used to investigate vasomotor responses against phenylephrine. Histopathology was used to determine vascular and cardiac remodeling processes, and mitochondrial health and functioning were assessed biochemically. Detailed methods can be found below.

### 4.2. Animals and Procedures

Animal care and experiments were approved by the Institutional Animal Care and Use Committee (CCD # AVD105002015129) and in accordance with the ARRIVE guidelines. Forty (40) male Wistar rats (Harlan, 7–9 weeks old, weighing 276 ± 15 g) were used. PAH was induced by a single i.p. injection of monocrotaline (60 mg·kg^−1^) and the placement of an aortocaval shunt one week later as described previously [54,55].

Rats were randomly assigned to four experimental groups: sham controls (*n* = 15), PAH (*n* = 5), PAH + vehicle (*n* = 10), and PAH + SUL-150 (*n* = 10). A subset of rats (sham, *n* = 5; PAH, *n* = 5) was sacrificed at day 14 post-shunting to assess the acute effects of SUL-150 in ex vivo myography of the pulmonary artery. The remaining rats (*n* = 10 per group) were followed for four weeks to study the effect of chronic SUL-150 administration.

The administration of vehicle and SUL-150 started 7 days post-monocrotaline injection and was performed by the i.p. implantation of osmotic minipumps (Alzet model #2004, Durect Corp., Cupertino, CA, USA) at the time of aortocaval shunt placement. The Lynch coil method was used to circumvent compatibility issues with the osmotic minipump reservoir as described previously [56]. Vehicle (dimethyl sulfoxide (DMSO) or SUL-150 ((*R*)-(6-hydroxy-2,5,7,8-tetramethylchroman-2-yl)(piperazin-1-yl)methanone, 0.9 M in DMSO, Sulfateq BV, Groningen, The Netherlands) was administered at a flow rate of 0.25 µL·h^−1^, accumulating to an approximate dose of 6 mg·kg^−1^·day^−1^ SUL-150 in a 300 g rat. Animals were weighed, watched for dyspnea, cyanosis, and oedema, and sacrificed preterm when a 15% weight loss or debilitating dyspnea occurred.

Three rats died during the study without a clear cause at necropsy and were excluded (sham control (*n* = 1) and vehicle-treated (*n* = 2) rats). At sacrifice, 7 rats were excluded from the study because of procedural motives: in two rats (vehicle (*n* = 1) and SUL-150-treated (*n* = 1) rats), adequate AC shunt could not be confirmed. In two other rats (both SUL-150-treated rats), the osmotic pumps were blocked by adipose deposits, limiting their function. Three rats died during cardiac function assessment (sham control (*n* = 2) and SUL-150 treated (*n* = 1) rats). Twenty rats were included in the study (sham control (*n* = 7), vehicle-treated (*n* = 7), and SUL-150-treated (*n* = 6) rats.

### 4.3. Echocardiogram and Hemodynamic Measurements

Echocardiographic (ECG) examination was performed under isoflurane anesthesia (5% in air during induction, 2% in air during maintenance) via inhalation prior to euthanisation using a Vivid Dimension 7 system and 10S-transducer (GE Healthcare, Waukesha, WI, USA). Ventricular dimensions, eccentricity index, and flow profiles over the aortic, pulmonary, and tricuspid valves were measured in standard views [57]. Cardiac output (mL·min^−1^) was calculated using the following equation: $CO({mL\cdot min}^{-1})=(\left(LVOTarea\right)\times \left(LVOTVTI\right))\times HR$, wherein LVOT area is the area of the left ventricular outflow tract, LVOT VTI is the velocity-time integral and HR is heart rate. Subsequent to ECG, invasive hemodynamic measurements were performed using a closed chest technique as described by others [58]. A fluid-filled pressure catheter was inserted into the right internal jugular vein and guided to the pulmonary artery under pressure waveform monitoring using a bedside monitor. Right ventricular systolic and diastolic pressure, pulmonary arterial pressures, and pulmonary wedge pressure were measured. Mean pulmonary arterial pressure (mmHg) was calculated as: mPAP(mmHg)=(23dPAP+13sPAP), wherein dPAP and sPAP are diastolic and systolic pulmonary artery pressure, respectively. Total pulmonary vascular resistance was estimated as: PVRmmHg·mL·min−1=(mPAP−PCWP)CO, wherein mPAP, mPWP, and CO are mean pulmonary artery pressure, pulmonary capillary wedge pressure, and cardiac output, respectively.

### 4.4. Organ Collection and Pathology

Rats were euthanized under 3–5% isoflurane anesthesia by exsanguination by aorta puncture. Hearts, lungs, and pulmonary arteries were harvested and processed for further analyses. The hearts were weighed and dissected into the left atrium, ventricle, right atrium, and ventricle and septum, which were all individually weighted. The Fulton Index was calculated as: Fulton Index=RV(LV+IVS), wherein RV and LV are right and left ventricular weights, respectively, and IVS is the weight of the interventricular septum. The right ventricle was cut in half, and one half was fixed in 3.6% formalin and embedded in paraffin. The other half was snap frozen in liquid nitrogen. The left lung lobe was fixed by filling the airways with 3.6% formalin and embedded in paraffin. The right lung lobe was snap frozen in liquid nitrogen.

### 4.5. Ex Vivo Myography

The fourth-degree branching intralobar pulmonary arteries were carefully dissected under a Leica S4E stereomicroscope as described before [59]. Arterial ring segments (2 mm length) were mounted in a Mulvany-type wire myograph system (model #610M, Danish Myo Technology, Aarhus, Denmark) using Ø40 µm steel wire. Rings were preserved in myography chambers filled with aerated Krebs buffer (pH 7.4) at 37 °C. After 20 min equilibration, rings were stretched to 1.8 kPa using the DMT Normalization Procedure and allowed to equilibrate for an additional 20 min. Analogue–digital signal conversion was performed using PowerLab 8/30, and a myogram was recorded in PowerLab Chart v 5.3 (AD Instruments, Oxford, UK). To assess the acute effects of SUL-150 on vasoconstriction, arterial rings were pre-constricted with 60 mM KCl and, following washout and stabilization, incubated with vehicle (0.1% DMSO) or SUL-150 (10^−4^ M) for 20 min. Next, constriction responses to phenylephrine (phenylephrine, Sigma-Aldrich, St. Louis, MO, USA) were recorded in the dose range of 3 × 10^−9^ to 3 × 10^−5^ M. To assess the chronic effects of SUL-150 treatment, a similar protocol was followed, leaving out the preincubation with vehicle or SUL-150. Isometric responses were calculated as % of maximal KCl-induced constriction, and dose–response curves were recorded to determine the efficacy (E_max_) and potency (EC_50_) of phenylephrine to induce vasoconstriction.

### 4.6. Histochemistry and Immunofluorescence

To assess cardiomyocyte hypertrophy, 4 µm thick right ventricular sections were stained using fluorescein-conjugated wheat germ agglutinin (ThermoFisher, Waltham, MA, USA) according to the manufacturer’s protocol, and >5 random images with perpendicularly cut bundles per rat were imaged at 40× magnification on an AxioObserver Z1 microscope in fluorescence mode (Zeiss, Jena, Germany). Cross-sectional cardiomyocyte area was determined for approximately 400 cardiomyocytes/sample (range 182–768) using Fiji 2.1.0/1.53c [60] in binary mode with the following particle analyzer settings: circularity 0.25–1.00 and pixel area 300–1000 px^2^.

Cardiac fibrosis was measured after picrosirius staining and counterstaining with Weigert’s hematoxylin (both Sigma-Aldrich, St. Louis, MO, USA) following the manufacturer’s instructions. Samples were imaged on a NanoZoomer digital slide scanner (Hammamatsu Photonics, Shizuoka, Japan) at 40× magnification. Interstitial fibrosis (i.e., non-perivascular fibrosis) was quantified using Aperio ImageScope (Leica Biosystems, Nussloch, Germany). Morphology of the intra-acinar pulmonary vessels was assessed after Verhoeff-Von Gieson staining (Sigma-Aldrich, St. Louis, MO, USA). Samples were imaged on a Leica DM2000 microscope at 63× magnification using Leica Image Acquisition software version 4.13 (Leica, Wetzlar, Germany). Pulmonary vessel remodeling was quantified as described previously [54,61].

### 4.7. mtDNA Copy Number, Radical Scavenging, Lipid Peroxidation and ATP Measurements

Right ventricular samples were homogenized in lysis buffer (100 mM NaCl, 10 mM EDTA, 0.5% SDS in 20 mM Tris-HCl, pH 7.4) containing 50 U·mL^−1^ RNase I and 100 U·mL^−1^ proteinase K (both Thermo Scientific, Waltham, MA, USA). After overnight incubation at 55 °C, total DNA was precipitated using 2-propanol. Aliquots of 5 ng total DNA were amplified on a ViiA7 Real-time PCR system (ThermoFisher, Waltham, MA, USA) using iTaq Universal SYBR Green Supermix (Bio-Rad, Hercules, CA, USA) and primers specific for mitochondrial DNA (MT-ND1; sense 5′-CCTCCTAATAAGCGGCTCCT-3′, antisense 5′-GGCGGGGATTAATAGTCAGA-3′) or nuclear DNA (NDUFA1; sense 5′-ATGGCCCGAACCAAGCAGACC-3′, antisense 5′-TTAAGCTCTCTCCCCCCGTATCCG-3′). MtDNA copy number was calculated as: $mtDNA=2\times 2^{Cq\left(NDUFA1\right)-Cq(MT-ND1)}$.

Right ventricular samples were homogenized in ddH_2_O using a TissueRuptor II (Qiagen, Hilden, Germany) followed by sonication at 20 kHz for 3 × 1 min (Sonopuls 2000, Bandelin, Berlin, Germany) and centrifugation at 14,000× *g* to separate insoluble proteins. Supernatant was used to assess radical scavenging activity by ABTS-radical decolorization as described by Re et al. [62], lipid peroxidation by assessing the reactivity to thiobarbituric acid according to Ohkawa et al. [63], and ATP content was determined using the ATP Determination Kit (ThermoFisher, Waltham, MA, USA) according to the manufacturer’s instructions. Total protein content of the supernatants was determined by DC Protein Assay (Bio-Rad, Hercules, CA, USA) and used for data normalization.

### 4.8. Immunoblotting

Protein lysates were prepared from the right ventricle using RIPA buffer (ThermoFisher, Waltham, MA, USA) and supplemented with 0.5% proteinase inhibitor cocktail (Sigma-Aldrich, St. Louis, MO, USA) and 0.5% Halt™ phosphatase inhibitor (ThermoFisher, Waltham, MA, USA). Total protein content of the lysates was determined by DC Protein Assay (Bio-Rad, Hercules, CA, USA). Twenty micrograms of total protein per sample was loaded on mini-PROTEAN precast gradient gels (4–15% denaturing SDS–polyacrylamide gel; Bio-Rad, Hercules, CA, USA), separated by gel electrophoresis, and blotted onto nitrocellulose membrane using the Trans-Blot turbo system (Bio-Rad, Hercules, CA, USA) according to standard protocols. Blots were blocked with 5% bovine serum albumin in 20 mM Tris-HCl (pH 7.4) at room temperature for 30 min and incubated at 4 °C overnight with primary antibodies to mtTFA (sc-166965, Santa Cruz Biotechnology, Dallas, TX, USA), PGC1α (NBP1-04676, Bio-Techne, Abingdon, UK), PINK1 (BC100-494, Bio-Techne, Abingdon, UK), Parkin (sc-32282, Santa Cruz Biotechnology, Dallas, TX, USA), or Tubulin (Sigma-Aldrich, St. Louis, MO, USA), all at a dilution of 1:500. Alkaline phosphatase-conjugated secondary antibodies and NBT/BCIP (Bio-Rad, VA, USA) were used for detection. Densitometric analysis was performed using Totallab 120 (Nonlinear Dynamics, Newcastle upon Tyne, UK).

### 4.9. Cell Culture and Hypertrophy Reporters

Cultures of neonatal rat cardiomyocytes were prepared from 1–3-day-old Wistar rats and adenovirally transformed with hypertrophy reporter construct as described elsewhere [64]. Cardiomyocytes were pre-incubated with SUL-150 (3 × 10^−5^ M) or vehicle for 30 min and subsequently stimulated with phenylephrine in the dose range of 10^−8^ to 10^−4^ M for 24 h. Cells were lysed in passive lysis buffer (Promega, Madison, WI, USA), and fluorescence intensity of the lysate was recorded in an EnVision 2102 plate reader (Perkin Elmer, Waltham, MA, USA).

### 4.10. Measurement of SUL-150 Levels in Plasma, Lung, and Right Ventricle

Measurement of SUL-150 levels in plasma and tissue homogenates was performed by liquid chromatography and mass spectrometry. Plasma or tissue homogenates were supplemented with acetonitrile and sonicated, followed by centrifugation at 14,000× *g* to liberate SUL-150 from the samples and pellet protein precipitates. Tissue samples were subjected to additional solid-phase extraction using an SPEStrata C-18 cartridge (100 mg, 55 µm, 70 Å, Phenomenex, Torrance, CA, USA), prepared with 1 mL methanol followed by 1 mL water. Analytes were washed by methanol (10%) and eluted in acetonitrile/methanol (3:7 *v*/*v*). Eluate was concentrated, and water (20%) was added. Analyte recovery was >95% for plasma samples and >70% for tissue. Liquid chromatography of the samples was performed on a 1260 Infinity HPLC device (Agilent Tech., Santa Clara, CA, USA) using a ZORBAX Eclipse AAA column (3.0 × 150 mm, particle size 3.5 µm) in a reversed-phase setup and a flow rate of 0.5 mL·min^−1^ with gradient elution. Solvents consisted of methanol (6%)/acetonitrile (4%) acetate and methanol (54%)/acetonitrile (36%) in water with 0.1% ammonium for solvents A and B, respectively. MS/MS detection was performed on a QQQ 6460 mass spectrometer (Agilent Tech., Santa Clara, CA, USA). Detection was set for a quantifier ion (205.1, CE 25 V) and qualifier ion (190.1, CE 40 V). Gas temperature for MS was set to 300 °C and gas flow was set to 6 L·min^−1^. Quantification of the samples was performed using an external standard for calibration. LOD and LOQ were 5 pg·mL^−1^ and 17 pg·mL^−1^, respectively.

### 4.11. Statistical Analyses

Continuous data are expressed as the mean ± S.D., and statistical inference was performed using ANOVA followed by Fisher LSD post hoc analyses. Discrete data are expressed as *n* and assessed by *χ*2 goodness-of-fit *t*-test followed by z-test post hoc analyses. Data from dose–response relationships were modeled by four-parameter logistic regression and statistically evaluated by F-test. Probabilities < 0.05 were considered statistically significant.

## 5. Conclusions

Here, we show that the novel 6-chromanol SUL-150 limits disease progression in PAH rats. SUL-150 ameliorates pulmonary vascular remodeling and right ventricular failure secondary to PAH by the reduction in pulmonary artery remodeling and concurrent pulmonary vascular resistance and increase in pulmonary perfusion. SUL-150 precludes the development of right ventricular mitochondrial dysfunction and oxidative stress secondary to experimental PAH and prevents deterioration of right ventricular contractility and cardiac output. Hence, SUL-150 may be considered a promising therapeutic candidate in the treatment of pulmonary arterial hypertension and secondary right ventricular failure.

## Figures and Tables

**Figure 1 ijms-26-07181-f001:**
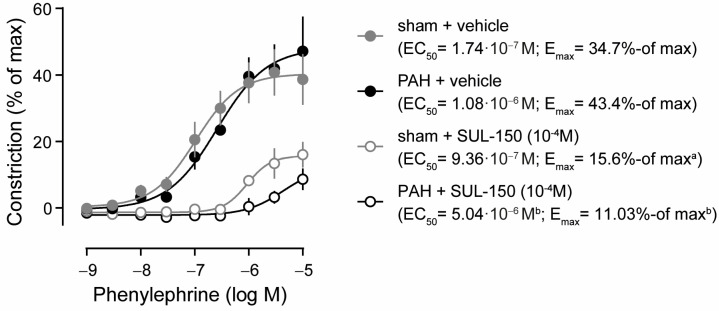
SUL-150 reduces phenylephrine-induced vasoconstriction of the pulmonary artery in experimental PAH (day 21 post-monocrotaline administration). Phenylephrine induces vasoconstriction of the pulmonary artery ex vivo in sham (grey dots) and PAH (black dots) rats with equipotency (EC_50_) and similar efficacy (E_max_). In the pulmonary arteries of sham rats, pre-incubation with 10^−4^ M SUL-150 for 30 min (grey open circles) decreases efficacy of vasoconstriction, but not the potency, as compared to vehicle control artery rings from sham rats (grey dots). In the pulmonary arteries of PAH rats, pre-incubation with SUL-150 (black open circles) decreases both the potency and efficacy of phenylephrine-induced vasoconstriction as compared to vehicle-treated artery rings (black dots). Data are expressed as mean ± S.D. of five rats per group (biological replicates). Experiments were performed in triplicate (technical replicates). Statistical evaluation was performed on individually determined EC_50_ and E_max_ values by ANOVA with Fisher’s LSD post hoc analyses. ^a^
*p* < 0.05 versus sham + vehicle; ^b^
*p* < 0.05 versus PAH + vehicle. Data are expressed as mean ± S.D. IVS = interventricular septum, LA = left atrium, LV = left ventricle, RA = right atrium, RV = right ventricle, RVP = right ventricular pressure at (d) end-diastole or (s) end-systole.

**Figure 2 ijms-26-07181-f002:**
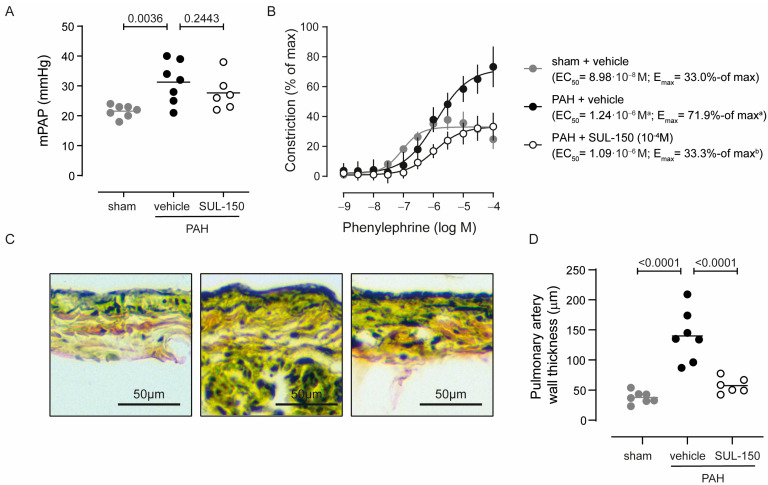
SUL-150 normalizes pulmonary artery pressure by reducing pulmonary artery remodeling (day 35 post-monocrotaline administration and treatment). Rats subjected to monocrotaline and aortocaval shunt placement develop pulmonary arterial hypertension as shown by an (**A**) increase in mean pulmonary arterial pressure, which is not affected by SUL-150 administration. *N* = 6–7 rats per group (plotted individually). Statistical evaluation was performed by ANOVA with Fisher’s LSD post hoc analyses. (**B**) In ex vivo pulmonary artery rings of PAH rats (black dots), phenylephrine-induced vasoconstriction with a higher potency and efficacy when compared to pulmonary artery rings from sham control rats (grey dots). SUL-150 administration in PAH rats (black open circles) normalized the efficacy of phenylephrine-induced vasoconstriction to the level of sham control rats. The potency of phenylephrine-induced vasoconstriction was unaltered in PAH rats that were administered with SUL-150 and had a similar potency to vehicle-treated rats. Data are expressed as mean ± S.D. of 6–7 rats per group (biological replicates) and were performed in triplicate (technical replicates). Statistical evaluation was performed on individually determined EC_50_ and E_max_ values by ANOVA with Fisher’s LSD post hoc analyses. (**C**,**D**) Pulmonary artery wall thickness is increased in vehicle-treated PAH rats as compared to vehicle-treated sham control rats, suggestive of outward remodeling of the pulmonary artery. Chronic SUL-150 administration in PAH rats, precludes pulmonary artery remodeling and the pulmonary artery wall thickness of SUL-150-treated PAH rats does not differ from sham control rats. *N* = 6–7 rats per group (plotted individually). Statistical evaluation was performed by ANOVA with Fisher’s LSD post hoc analyses. ^a^
*p* < 0.05 versus sham + vehicle; ^b^
*p* < 0.05 versus PAH + vehicle.

**Figure 3 ijms-26-07181-f003:**
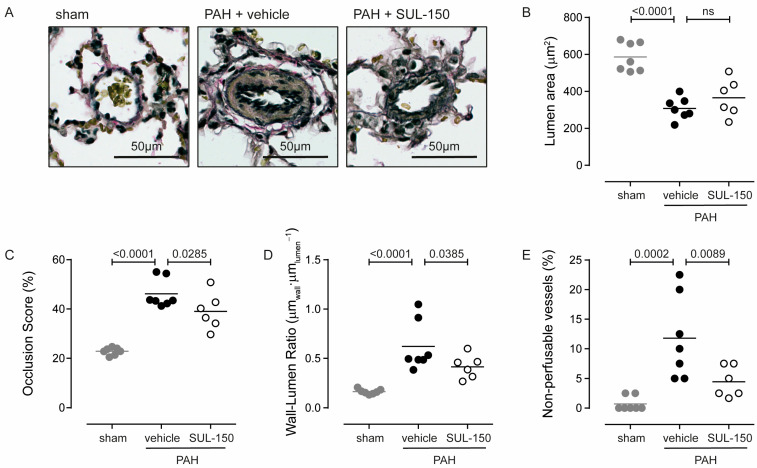
SUL-150 reduces pulmonary vessel remodeling in PAH. Rats subjected to monocrotaline and aortocaval shunt placement develop (**A**) right pulmonary vessel remodeling as indicated by (**B**) a decreasing lumen area and increasing (**C**) occlusion score and (**D**) wall–lumen ratio. SUL-150 administration reduces these pathological changes. In vehicle-treated PAH rats, (**E**) the number of non-perfusable vessels (vessels Ø < 10 µm^2^) is increased as compared to sham control rats. SUL-150 administration decreases the number of non-perfusable vessels. *N* = 6–7 rats per group (plotted individually). Statistical evaluation was performed by ANOVA with Fisher’s LSD post hoc analyses. ns = non-significant.

**Figure 4 ijms-26-07181-f004:**
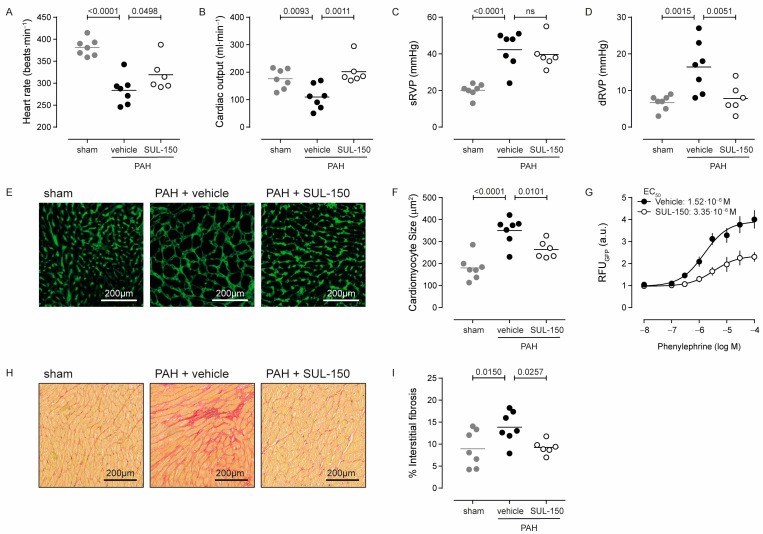
SUL-150 mitigates right ventricular failure in PAH. Rats subjected to monocrotaline and aortocaval shunt placement develop right ventricular failure as indicated by a decrease in (**A**) heart rate and (**B**) cardiac output and increased (**C**) systolic right ventricular pressure and (**D**) diastolic right ventricular pressure. SUL-150 administration ameliorates these pathological changes to right ventricular function. Right ventricular failure in vehicle-treated PAH rats is associated with (**E**,**F**) cardiomyocyte hypertrophy, which is mitigated by SUL-150 administration. *N* = 6–7 rats per group (plotted individually). Statistical evaluation was performed by ANOVA with Fisher’s LSD post hoc analyses. (**G**) In vitro, SUL-150 dose-dependently decreases the potency and efficacy of phenylephrine to produce fluorescence by neonatal rat cardiomyocytes transduced with a virus encoding for a hypertrophy reporter. Data are expressed as mean ± S.D. of 4 biological replicates. Statistical evaluation was performed on individually determined EC_50_ and E_max_ values by ANOVA with Fisher’s LSD post hoc analyses. Furthermore, (**H**,**I**) right ventricular failure associates with increased fibrogenesis in vehicle-treated PAH rats, which is mitigated by SUL-150 administration. *N* = 6–7 rats per group (plotted individually). Statistical evaluation was performed by ANOVA with Fisher’s LSD post hoc analyses.

**Figure 5 ijms-26-07181-f005:**
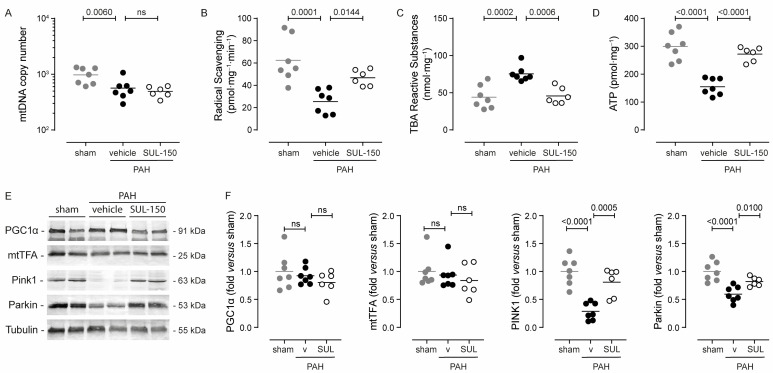
SUL-150 mitigates mitochondrial dysfunction in the right ventricle. Rats subjected to monocrotaline and aortocaval shunt placement show mitochondrial loss and the development of mitochondrial dysfunction as shown by a (**A**) decrease in mitochondrial DNA content, (**B**) radical scavenging capacity, and (**C**) increased lipid peroxidation levels, culminating in (**D**) a reduction in ATP content. SUL-150 administration mitigates these changes in PAH rats. *N* = 6–7 rats per group (plotted individually). Experiments were performed in triplicate (technical replicates). Statistical evaluation was performed by ANOVA with Fisher’s LSD post hoc analyses. (**E**,**F**) Immunoblotting for molecular markers of mitogenesis (PGC1α and mtTFA) and mitophagy (PINK1 and Parkin) reveals that in vehicle-treated PAH rats, the expression of PINK1 and Parkin is reduced, indicative of disturbed mitophagy. SUL-150 administration to PAH rats maintains the expression of PINK1 and Parkin. *N* = 6–7 rats per group (plotted individually). Statistical evaluation was performed by ANOVA with Fisher’s LSD post hoc analyses.

**Table 1 ijms-26-07181-t001:** In-life data during pulmonary arterial hypertension (PAH) development.

	Sham (*n* = 5)	PAH (*n* = 4)
Body weight		
Start weight (day 0) (g)	279.4 ± 11.30	271.3 ± 15.13
End weight (day 20–22) (g)	338.2 ± 8.14	303.0 ± 21.34 *
Weight change (%)	21.12 ± 2.55	11.68 ± 3.72 *
Pulmonary blood pressure (mmHg)		
Mean	20.20 ± 3.31	31.89 ± 1.84 *
Systolic	24.20 ± 3.96	36.33 ± 1.52 *
Diastolic	18.20 ± 3.11	29.67 ± 2.08 *
Right ventricular pressure (mmHg)		
sRVP	19.00 ± 2.55	27.75 ± 8.18 *
dRVP	12.50 ± 4.74	13.50 ± 5.45
Heart weight (relative to body weight)		
Heart weight (mg·g^−1^)	2.63 ± 0.12	4.0 ± 0.40 *
LA weight (mg·g^−1^)	0.10 ± 0.04	0.18 ± 0.03 *
LV weight (mg·g^−1^)	1.13 ± 0.09	1.63 ± 0.20 *
RA weight (mg·g^−1^)	0.11 ± 0.03	0.25 ± 0.12
RV weight (mg·g^−1^)	0.61 ± 0.06	1.14 ± 0.11 *
IVS Weight (mg·g^−1^)	1.13 ± 0.09	1.63 ± 0.20 *
Fulton Index	0.34 ± 0.05	0.46 ± 0.06 *

Data are expressed as mean ± S.D. IVS = interventricular septum, LA = left atrium, LV = left ventricle, RA = right atrium, RV = right ventricle, RVP = right ventricular pressure at (d) end-diastole or (s) end-systole. * *p* < 0.05 vs. sham.

**Table 2 ijms-26-07181-t002:** SUL-150 concentration in plasma, lung, and heart.

		SUL-150	
Animal Number	Plasma (ng·mL^−1^)	Lung Tissue (pg·mg^−1^)	Right Ventricle (pg·mg^−1^)
E2R3	93.16	35.56	13.49
E2R11	3.31	10.21	29.91
E2R19	18.36	14.26	23.67
E2R23	4.42	25.74	16.80
E2R27	18.81	26.09	21.76
E2R31	43.90	53.94	143.68

**Table 3 ijms-26-07181-t003:** Hemodynamic parameters.

	PAH		
	Sham (*n* = 7)	Vehicle (*n* = 7)	SUL-150 (*n* = 6)
Pulmonary artery pressure (mmHg)			
Mean	21.6 ± 2.1	31.3 ± 7.1 ^a^	27.7 ± 5.8 ^a^
Systolic	27.0 ± 1.6	41.4 ± 9.3 ^a^	37.7 ± 8.7 ^a^
Diastolic	14.8 ± 3.3	25.4 ± 6.7 ^a^	19.0 ± 5.7
Pulmonary wedge pressure (mmHg)			
Mean	7.2 ± 4.2	8.2 ± 3.8	4.8 ± 1.9
Vascular resistance (mmHg·mL·min^−1^)			
Pulmonary	0.08 ± 0.02	0.30 ± 0.16 ^a^	0.12 ± 0.05 ^b^

Data are expressed as mean ± S.D. ^a^ *p* < 0.05 versus sham, ^b^ *p* < 0.05 versus vehicle.

**Table 4 ijms-26-07181-t004:** In-life data during chronic SUL-150 administration.

	PAH		
	Sham (*n* = 7)	Vehicle (*n* = 7)	SUL-150 (*n* = 6)
Body weight			
Start weight (day 0) (g)	272.6 ± 15.5	276.9 ±14.7	277.8 ± 15.4
End weight (day 32–33) (g)	358.0 ± 17.5	318.7 ± 16.6 ^a^	319.3 ± 12.0 ^a^
Weight change (%)	31.5 ± 6.6	15.2 ± 5.0 ^a^	15.1 ± 4.1 ^a^
Discomfort frequency			
Dyspnea (% of rats)	0	87.3 ^a^	44.4 ^a,b^
Cyanosis (% of rats)	0	14.3 ^a^	11.1 ^a^
Edema (% of rats)	0	14.3 ^a^	0 ^a,b^
Survival			
Mortality (n/%)	1 (12.5%)	2 (22.2%)	0 (0%) ^a,b^

Continuous data are expressed as mean ± S.D.; discrete data are expressed as percentage of rats or *n*. ^a^ *p* < 0.05 versus sham, ^b^ *p* < 0.05 versus PAH + vehicle.

**Table 5 ijms-26-07181-t005:** Cardiac parameters.

	PAH		
	Sham (*n* = 7)	Vehicle (*n* = 7)	SUL-150 (*n* = 6)
Heart weight (relative to body weight)			
Heart weight (mg·g^−1^)	2.61 ± 0.09	4.65 ± 0.25 ^a^	4.37 ± 0.66 ^a^
LA weight (mg·g^−1^)	0.10 ± 0.03	0.19 ± 0.04 ^a^	0.23 ± 0.11 ^a^
LV weight (mg·g^−1^)	1.18 ± 0.06	1.69 ± 0.22 ^a^	1.60 ± 0.29 ^a^
RA weight (mg·g^−1^)	0.12 ± 0.03	0.48 ± 0.10 ^a^	0.38 ± 0.12 ^a^
RV weight (mg·g^−1^)	0.55 ± 0.05	1.42 ± 0.14 ^a^	1.23 ± 0.31 ^a^
IVS weight (mg·g^−1^)	0.67 ± 0.07	0.85 ± 0.14	0.81 ± 0.22
Fulton Index	0.30 ± 0.03	0.56 ± 0.07 ^a^	0.51 ± 0.09 ^a^
Cardiac dimensions			
Ventricular Dimensions			
LVIDd (mm)	5.00 ± 0.67	4.77 ± 1.31	4.27 ± 0.46
RVIDd (mm)	3.40 ± 0.72	6.18 ± 1.59 ^a^	3.87 ± 1.37 ^b^
RVIDd/LVIDd ratio	0.49 ± 0.34	1.41 ± 0.59 ^a^	0.92 ± 0.35
Eccentricity Index			
Systole	0.90 ± 0.06	0.65 ± 0.14 ^a^	0.75 ± 0.07 ^a^
Diastole	0.95 ± 0.03	0.64 ± 0.13 ^a^	0.79 ± 0.13 ^a,b^
Ventricular Function			
Left ventricle			
LVOT diameter (mm)	3.44 ± 0.24	3.10 ± 0.16 ^a^	3.37 ± 0.24 ^b^
LVOT Vmax (m·s^−1^)	1.02 ± 0.18	1.06 ± 0.34	1.12 ± 0.39
LVOT Pmax (mmHg)	4.24 ± 1.44	4.91 ± 2.72	5.57 ± 3.73
LVOT VTI (mm)	5.10 ± 1.15	6.17 ± 1.86	6.16 ± 2.33
Right ventricle			
TAPSE (mm)	2.56 ± 0.38	2.24 ± 0.51	2.90 ± 0.49 ^b^
PV Vmax (m·s^−1^)	1.20 ± 0.15	1.18 ± 0.21	1.14 ± 0.19
PV Pmax (mmHg)	6.01 ± 1.17	5.51 ± 1.88	5.74 ± 1.50
PAAT (ms)	22.11 ± 6.58	13.86 ± 4.64 ^a^	17.07 ± 4.17
PV Acceleration slope (m·s^2^)	57.37 ± 17.88	95.19 ± 37.92 ^a^	65.37 ± 19.75 ^b^
Tricuspid valve insufficiency			
Average grade (0–3)	0.0 ± 0.0	2.43 ± 0.53 ^a^	1.0 ± 1.0 ^a,b^
None (%)	100	0	42.9
Mild (%)	0	0	14.3
Moderate (%)	0	57.1	42.9
Severe (%)	0	42.9	0

BW = body weight, LA = left atrium, LV = left ventricle, RA = right atrium, RV = right ventricle, IVS = interventricular septum, LVIDd = left ventricle inner diameter in diastole, RVIDd = right ventricle inner diameter in diastole, LVOT = left ventricle outflow tract, V_max_ = maximal flow speed, P_max_ = maximal pressure, VTI = velocity-time integral, TAPSE = tricuspid annular plane systolic excursion, PV = pulmonary valve, PAAT = pulmonary artery acceleration time. Data are expressed as mean ± S.D. ^a^ *p* < 0.05 versus sham, ^b^ *p* < 0.05 versus PAH + vehicle.

## Data Availability

All data relevant to this manuscript have been included in this manuscript. Raw data, reagents, and research procedures can be obtained from the corresponding author upon reasonable request.

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
