# Peer review of "SUL-150 Limits Vascular Remodeling and Ventricular Failure in Pulmonary Arterial Hypertension"

_ijms, 2025, doi:10.3390/ijms26157181_

Round 1
Reviewer 1 Report
Comments and Suggestions for Authors
The manuscript by Jorna and colleagues entitled SUL-150 limits vascular remodeling and ventricular failure in pulmonary arterial hypertension. PAH leads to progressive pulmonary vascular remodeling and RV failure. This study induced PAH in rats using monocrotaline and an aortocaval shunt, followed by treatment with the mitoprotective compound SUL-150. Treatment with the mitoprotective agent SUL-150 improved pulmonary vascular remodeling, prevented RV hypertrophy and fibrosis, and alleviated dyspnea. Despite not preventing mitochondrial loss, SUL-150 enhanced mitochondrial health, preserving RV function. Although it shows some interesting results, several points regarding the experimental design need to be addressed. The following major suggestions are for the authors and editors' considerations. Some specific comments are as follows:
Major:
- Figure 1, phenylephrine-induced vasoconstriction of the pulmonary artery is similar in sham and PAH group (vehicle). However, in figure 2B, PAH was mor likely to contract than the sham group. Please explain.
- Line125-126, more details should be given on the descript (average value??)
- The study uses a single dose of SUL-150 (~6 mg.kg-1.day-1) administered intraperitoneally via osmotic minipump but does not justify this dose selection. The absence of pharmacokinetic and bioavailability data leaves uncertainty regarding the optimality of this dose, as it is unclear whether higher or lower doses might yield comparable or improved effects. Dose-response studies and plasma/tissue drug concentration measurements would help establish therapeutic relevance Is it this dosage is optimal or if different routes of administration might be more effective? Please explain.
- The analysis reveals significant improvements in pulmonary artery remodeling and a reduction in vascular resistance. However, it fails to elucidate whether SUL-150 directly affects vascular cells, such as endothelial cells, smooth muscle cells, and fibroblasts, or if the observed benefits are largely attributable to enhanced right ventricular function.
- SUL-150 is recommended as a potential therapeutic, but how does it compare to standard PAH treatments (e.g., prostacyclin analogs, PDE5 inhibitors, endothelin receptor antagonists)? The inclusion of a comparison group utilizing standard PAH treatments, whether administered alone or alongside SUL-150, would provide greater translational relevance to the research.
- The analysis does not mention the possible systemic side effects associated with SUL-150, which are critical for its transition into clinical use. It is important to examine liver/kidney toxicity markers and systemic hemodynamics should be assessed to ensure safety.
- Only male rats were used in this study. Considering the pronounced sex bias associated with PAH, which is more prevalent in females, it is important to assess whether the effects of SUL-150 differ based on sex.
- Although SUL-150 improves mitochondrial health and prevents ventricular fibrogenesis, however, the exact molecular pathways involved are still not well defined. Conducting more in-depth mechanistic studies could enhance the findings of this research, such as mitochondrial dynamics (fusion/fission proteins like MFN2, OPA1, DRP1), fibrosis pathways (TGF-β/SMAD signaling, collagen deposition) or apoptosis and cell survival markers in both pulmonary and cardiac tissue.
The English could be improved to more clearly express the research.
Author Response
Referee 1
The manuscript by Jorna and colleagues entitled SUL-150 limits vascular remodeling and ventricular failure in pulmonary arterial hypertension. PAH leads to progressive pulmonary vascular remodeling and RV failure. This study induced PAH in rats using monocrotaline and an aortocaval shunt, followed by treatment with the mitoprotective compound SUL-150. Treatment with the mitoprotective agent SUL-150 improved pulmonary vascular remodeling, prevented RV hypertrophy and fibrosis, and alleviated dyspnea. Despite not preventing mitochondrial loss, SUL-150 enhanced mitochondrial health, preserving RV function. Although it shows some interesting results, several points regarding the experimental design need to be addressed. The following major suggestions are for the authors and editors' considerations. Some specific comments are as follows:
A: We thank the referee for the critical and constructive analysis of our manuscript and his/her suggestions for improvements. The specific replies to the comments can be found below.
- Figure 1, phenylephrine-induced vasoconstriction of the pulmonary artery is similar in sham and PAH group (vehicle). However, in figure 2B, PAH was mor likely to contract than the sham group. Please explain.
A1: We thank the reviewer for his/her question and the possible confusion on the difference in response in the PAH group between figs 1 and 2. For the data described in figure 1, the rats were sacrificed at day 21 to assess the acute effects of SUL-150 on vasomotor responses. In contrast, for the data described in figure 2, the rats were sacrificed at the end of the experiment (day 35) to assess the effects of chronic SUL-150 therapy. Thus, the difference in response observed here is derived from the progressive remodeling of the pulmonary artery in PAH.
Although this difference was detailed in the method section of our manuscript, to further clarify this difference and avoid confusion, we have specified the timing of the experiment in the figure legend and manuscript text in our revised manuscript.
- Line125-126, more details should be given on the descript (average value??)
A2: We apologize for this omission. Indeed, the values in lines 125-126 reflect group average values for plasma, lung and right ventricle concentrations of SUL-150 of which the individual levels per animal can be found in Table 2. We have corrected this omission in the revised manuscript.
- The study uses a single dose of SUL-150 (~6 mg.kg-1.day-1) administered intraperitoneally via osmotic minipump but does not justify this dose selection. The absence of pharmacokinetic and bioavailability data leaves uncertainty regarding the optimality of this dose, as it is unclear whether higher or lower doses might yield comparable or improved effects. Dose-response studies and plasma/tissue drug concentration measurements would help establish therapeutic relevance Is it this dosage is optimal or if different routes of administration might be more effective? Please explain.
A3: We fully agree with the reviewer that studies including dose-response relationships on potential therapeutic compounds further establish the therapeutic relevance and facilitates further clinical development. Yet, the current manuscript discusses the first “proof-of-concept” that the pharmacological targeting of mitochondrial function has therapeutic merit in PAH. Hence, we chose to administer the highest achievable dose of the compound, which was determined by its solubility in DMSO and the characteristics (i.e. flow) of the minipump. Given that SUL-150 treatment has a positive effect on RV function, in ongoing and future studies we will expand on the pharmacokinetic profile of the compound in this model. We have added this rationale to a new “limitations” section in out revised manuscript.
- The analysis reveals significant improvements in pulmonary artery remodeling and a reduction in vascular resistance. However, it fails to elucidate whether SUL-150 directly affects vascular cells, such as endothelial cells, smooth muscle cells, and fibroblasts, or if the observed benefits are largely attributable to enhanced right ventricular function.
A4: We agree with the reviewer that in the present study we cannot elucidate if the improvements in pulmonary artery remodeling and the reduction in vascular resistance precluded right ventricular failure or that the benefits are attributable to a direct effect on ventricular cardiomyocytes. Moreover, these three effects may be synergistic in the amelioration of the observed cardiac phenotype. Given the systemic actions of SUL-150 it would be impossible to isolate the individual effects without pharmacologically antagonizing specific responses. We have discussed these confounders in the discussion.
- SUL-150 is recommended as a potential therapeutic, but how does it compare to standard PAH treatments (e.g., prostacyclin analogs, PDE5 inhibitors, endothelin receptor antagonists)? The inclusion of a comparison group utilizing standard PAH treatments, whether administered alone or alongside SUL-150, would provide greater translational relevance to the research.
A5: This is indeed a limitation of the current proof-of-concept study and a focus of future studies. Current PAH treatments are not curative and focused on symptom relief. Hence, exploring if SUL-150 would provide better therapeutic outcomes alone or on top of standard care is highly relevant. We have added the lack of a comparative treatment as a limitation of our study.
- The analysis does not mention the possible systemic side effects associated with SUL-150, which are critical for its transition into clinical use. It is important to examine liver/kidney toxicity markers and systemic hemodynamics should be assessed to ensure safety.
A6: We agree with the referee that for further (pre)clinical development of SUL-150 a full toxicokinetic profile of SUL-150 needs to be generated. Yet, at the present stage of development of the compound, this has not been a primary objective as first the therapeutic efficacy of SUL-150 was addressed. A comparable compound – i.e. SUL-138 – has been assessed for toxicokinetic responses in rats and minipigs and gave no indication of liver or kidney toxicity at dosages up to 136 mg/kg/day [Swart et al. 2024 Toxicol Rep].
SUL-150 is a vasodilator [Nakladal 2019 Sci Rep], which might influence its safety profile. Yet, in the current study no systemic hemodynamic effects – i.e. change in mean arterial pressure – were observed for the used dose level. These data are discussed in the manuscript.
- Only male rats were used in this study. Considering the pronounced sex bias associated with PAH, which is more prevalent in females, it is important to assess whether the effects of SUL-150 differ based on sex.
A7: We agree with the referee that there is sex dimorphism in PAH. Indeed, PAH develops more often in females, yet its progression is more aggressive in males who have a higher mortality rate. Therefore, we believe that the results obtained in our study are relevant to the PAH patient population, although we do believe that a potential sex difference in response to SUL-150 therapy should be investigated in future studies. We have listed the use of only male rats as a limitation to our study.
- Although SUL-150 improves mitochondrial health and prevents ventricular fibrogenesis, however, the exact molecular pathways involved are still not well defined. Conducting more in-depth mechanistic studies could enhance the findings of this research, such as mitochondrial dynamics (fusion/fission proteins like MFN2, OPA1, DRP1), fibrosis pathways (TGF-β/SMAD signaling, collagen deposition) or apoptosis and cell survival markers in both pulmonary and cardiac tissue.
A8: Uncovering the molecular mechanism of action was outside the scope of the current proof-of-concept study but will be explored in more detail in follow-up studies. Here, we sought to establish a rationale for mitochondrial therapy in PAH, which forms the basis for further research in which the molecular mechanisms underpinning a therapeutic response will be further explored. We have added this rational to our revised manuscript.
Reviewer 2 Report
Comments and Suggestions for Authors
Pulmonary hypertension is a major worldwide morbidity and mortality problem. Primitive pulmonary hypertension, in particular, has an extremely severe mortality and prognosis, affecting mainly young women. From this point of view, the authors have set an extremely interesting goal, with possible implications in cardiology practice.
Mitochondrial dysfunction plays an important role in the pathogenesis and development of pulmonary arterial hypertension. The use of a mitoprotective compound (SUL-150) to limit pulmonary artery endothelial remodeling may revolutionize the treatment of pulmonary hypertension, although the results are only experimental, on a small number of rats, they are still very encouraging.
The study is well conducted, the methodology is well described, meeting the rigors of a valuable research. The use of figures and tables increases the relevance and the degree of perception of the results.
I suggest that the authors to specify both in the abstract and in the article in extenso the aim of the study, because it is very important that the readers understand from the first lines the special relevance of the research.
The references are vast and up-to-date.
I encourage the authors to continue the research.
Author Response
Referee 2
Pulmonary hypertension is a major worldwide morbidity and mortality problem. Primitive pulmonary hypertension, in particular, has an extremely severe mortality and prognosis, affecting mainly young women. From this point of view, the authors have set an extremely interesting goal, with possible implications in cardiology practice.
Mitochondrial dysfunction plays an important role in the pathogenesis and development of pulmonary arterial hypertension. The use of a mitoprotective compound (SUL-150) to limit pulmonary artery endothelial remodeling may revolutionize the treatment of pulmonary hypertension, although the results are only experimental, on a small number of rats, they are still very encouraging.
The study is well conducted, the methodology is well described, meeting the rigors of a valuable research. The use of figures and tables increases the relevance and the degree of perception of the results.
I suggest that the authors to specify both in the abstract and in the article in extenso the aim of the study, because it is very important that the readers understand from the first lines the special relevance of the research.
The references are vast and up-to-date.
I encourage the authors to continue the research.
A: We thank the referee for this highly positive feedback. Indeed, PAH is a very severe condition that results in high morbidity and mortality for which no effective therapies are available to date. As per referee’s suggestion, we extended the description of our study aim – to show proof-of-concept for mitoprotective therapies in PAH) – in the revised manuscript. We take the positive feedback of the referee as an encouragement to continue our work on developing this compound into a clinically available therapy in future perspective.